# A data-driven analysis of spatiotemporal cues and experience accumulation effects for pitch type prediction

Ryota Takamido[1]*, Chiharu Suzuki[1], Hiroki Nakamoto[2]

**1** Sports Innovation Organization, National Institute of Fitness and Sports in Kanoya, Kanoya, Kagoshima, Japan, **2** Faculty of Physical Education, National Institute of Fitness and Sports in Kanoya, Kanoya, Kagoshima, Japan

* rtakamido@nifs-k.ac.jp

## Abstract

Conventional sports anticipation studies primarily rely on hypothesis-testing paradigms that target predetermined cues. However, such approaches risk overlooking unanticipated sources of predictive information. This study addresses this limitation by introducing a data-driven analysis using machine learning (ML) models as a complementary approach to conventional experimental research. Given that predictive cues embedded within movements can enhance the prediction accuracy of ML models, the proposed analysis identified spatiotemporal cues for prediction and quantified the effects of accumulating opponent-specific information across trials. Motion-capture data were collected from eight collegiate baseball pitchers, and joint-angle time series were analyzed using logistic regression models to predict pitch type (fastball vs. breaking ball). Specifically, two analyses were conducted: (1) a sliding time-window analysis to identify when and where predictive cues emerged within target motions and (2) a set-size analysis to evaluate how prediction accuracy varied with dataset size. The main results revealed that (1) predictive cues were distributed across the entire body, but models integrating whole-body information achieved the highest accuracy; (2) informative cues emerged in most body regions around the initiation of the pitcher's weight shift; (3) the accumulation of opponent-specific information had a pronounced effect up to approximately 30 pitches; and (4) substantial individual differences existed in when and which cues were effective for pitch-type prediction. These results clarify the similarities and differences between cues employed by human athletes and those utilized by ML models, thereby providing insights into athlete-specific cognitive strategies. Although alignment with human athletes must be carefully examined in future, a key theoretical contribution of this study is that it explores a complementary approach to conventional hypothesis-testing experiments by offering a time-resolved, data-driven account of where and when pitch-type–predictive information emerges in pitching kinematics.

**Data availability statement:** All data and code supporting the findings of this study are available on GitHub at https://github.com/takamido/Pitch_type_pred_ML.

**Funding:** This work was supported by the Japan Society for the Promotion of Science (Grant number: JP25K21018). The funders had no role in study design, data collection and analysis, decision to publish, or preparation of the manuscript.

**Competing interests:** The authors have declared that no competing interests exist.

# 1 Introduction

Anticipation—defined as the cognitive ability to observe others' movements, infer their intentions and goals, and predict forthcoming events—is a critical factor in athletic performance in dynamic, high-speed sports [1]. Given that the combined time required for perception, decision-making, and motor execution often exceeds the time available to athletes (e.g., the ball's flight time in hitting), anticipating future events and initiating actions in advance are essential for achieving task goals [2].

From a methodological perspective, numerous studies have investigated experts' anticipation skills using controlled experimental paradigms designed for hypothesis testing. In these studies, representative visual stimuli—such as an opponent's pitching motion or early ball-flight information in baseball hitting [3,4]—were presented to participants under specific experimental manipulations (e.g., occluding a particular body part). Participants' verbal or motor responses were collected and statistically analyzed to test the relevant hypotheses. Consequently, several unique characteristics underlying experts' superior anticipation skills have been identified, including their ability to utilize temporally advanced kinematic information from opponents [5,6], spatially localized cues from specific body parts [7,8], and individual differences associated with anticipatory ability [9,10].

However, a fundamental limitation of conventional sports anticipation research is its dependence on pre-established hypotheses and the risk of omitting relevant cues, as most studies have focused heavily on hypothesis-testing experimental designs. Because most previous studies selected candidate cues presumed to contribute to predictions and manipulated them experimentally, the main findings have largely been restricted to confirming or rejecting proposed hypotheses. Furthermore, if accurate prediction is achieved with only a limited number of spatiotemporal cues, participants may not need to use all available information, creating a risk that cues not commonly used across the studied group will be overlooked (Fig 1a). To achieve a comprehensive understanding of athletes' anticipation skills, it is necessary to clarify (1) the types of spatiotemporal prediction cues potentially embedded in an opponent's movement and (2) which of these cues athletes selectively use. However, conventional hypothesis-testing experiments primarily address the latter, making it difficult to systematically examine the former.

Addressing the shortcomings of conventional methods, data-driven analyses based on machine learning (ML) models have recently gained attention as a complementary approach [11–13]. These studies analyzed measured data using ML models to identify new insights, models, and patterns (e.g., [14,15]). Specifically, in the context of sports anticipation, we argue that ML models can help identify potential cues that enhance athletes' predictive skills and the effects of their accumulation across trials (Fig 1b). From an information-theoretic perspective, the information embedded in an opponent's movements should, in principle, be quantifiable directly from motion data by assessing how its use improves prediction accuracy—that is, reduces uncertainty. Similarly, the accumulation of opponent-specific information across trials should be quantifiable by evaluating how much it promotes the separation of

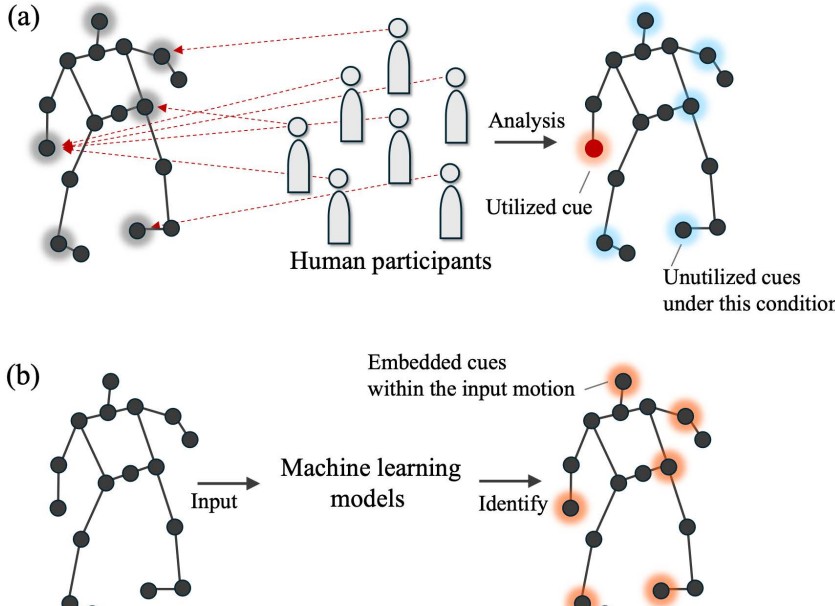

**Fig 1. A schematic comparison between (a) a hypothesis-testing experimental paradigm and (b) a data-driven machine learning analysis.**

informative cues from noise. These data-driven insights can deepen our understanding of sports anticipation skills by complementing conventional hypothesis-driven experiments.

Therefore, we aimed to develop a novel ML-based analysis that provides data-driven insights into athletes' anticipation skills. Specifically, we assumed that motion information improving ML prediction accuracy, by reducing uncertainty, can be regarded as a potential cue contributing to anticipation skills. Based on this assumption, we conducted two analyses: (1) identifying spatiotemporal cues for the ML model that enhance anticipation skills and (2) evaluating how accumulated opponent-specific information affects prediction accuracy of the model.

Although some recent studies have applied ML models to predict sports movement outcomes [16–19], these approaches have several limitations, including limited explainability [18], restriction of identified cues to spatial (body-part) information [19], and reliance on non-athlete motion data [16,17]. Unlike previous approaches, our method incorporates athletes' motion data and identifies interpretable spatiotemporal cues while providing a novel analysis of how accumulating opponent-specific information across trials contributes to improved ML's prediction accuracy. These findings provide valuable insights for advancing the application of ML techniques in sports science, particularly in contexts involving small datasets and high inter-individual variability.

## 2 Materials and methods

In developing the proposed ML analysis, this study selected pitch-type prediction in baseball (fastball vs. breaking ball) as the target task, given the complexity of the motion information provided by the opponent (i.e., the pitcher) and the importance of accurate prediction in coping with strict temporal constraints during real-game striking actions [2]. For the analysis, we constructed individual ML models for each pitcher, analogous to the way participants in experimental studies typically learn predictive models based on the actions of a single actor. This design also reflects real competitive contexts in which batters adapt to individual pitchers by progressively accumulating information across pitches. The source code and dataset used for the analysis are publicly available on GitHub (https://github.com/takamido/Pitch_type_pred_ML).

### 2.1 Overview of the proposed analysis

**2.1.1 Conceptual framework of the proposed analysis.** This analysis was inspired by recent neuroscience research that aimed to understand the spatiotemporal characteristics of electroencephalography (EEG) signals by validating the decoding accuracy of ML models [20,21]. In this study, we extended this framework to identify potential spatiotemporal cues in sports anticipation embedded within complex athletic movements and to evaluate how the accumulation of these cues across trials influences the prediction accuracy of the ML model. Fig 2 depicts a general framework for training and testing a pitch-type classification model using ML.

Conceptually, the task of the ML model is to decode the spatiotemporal motion data—defined in this study as a time series of joint angles—by leveraging the information contained in the training data to determine whether it corresponds to a fastball or breaking ball (Fig 2). If the model achieves high prediction accuracy using the given information, it can be inferred that the corresponding spatiotemporal features may contribute to improved prediction performance.

**2.1.2 The two ML analyses in the context of sports anticipation.** Building on this property of ML, two analyses were conducted in this study. Fig 3 presents a schematic image of the analysis design.

In Analysis 1, we used data-driven ML analysis to determine when and where the information for pitch-type classification was embedded within the spatiotemporal motion data (Fig 3a). In this analysis, we repeatedly trained the ML model using sliding time windows and evaluated its prediction accuracy, enabling the continuous characterization of temporal changes in the prediction probability. Furthermore, restricting the input to specific joints (e.g., those in the throwing arm) allowed for the identification of the joints and time points at which their information contributed to improving the prediction accuracy of the ML model. The prediction results obtained from the individual pitcher models were aggregated and statistically analyzed to (1) verify the presence of spatiotemporal cues for the ML model shared across individuals, and (2) identify ML design components and individual-specific factors that influence prediction accuracy of the model.

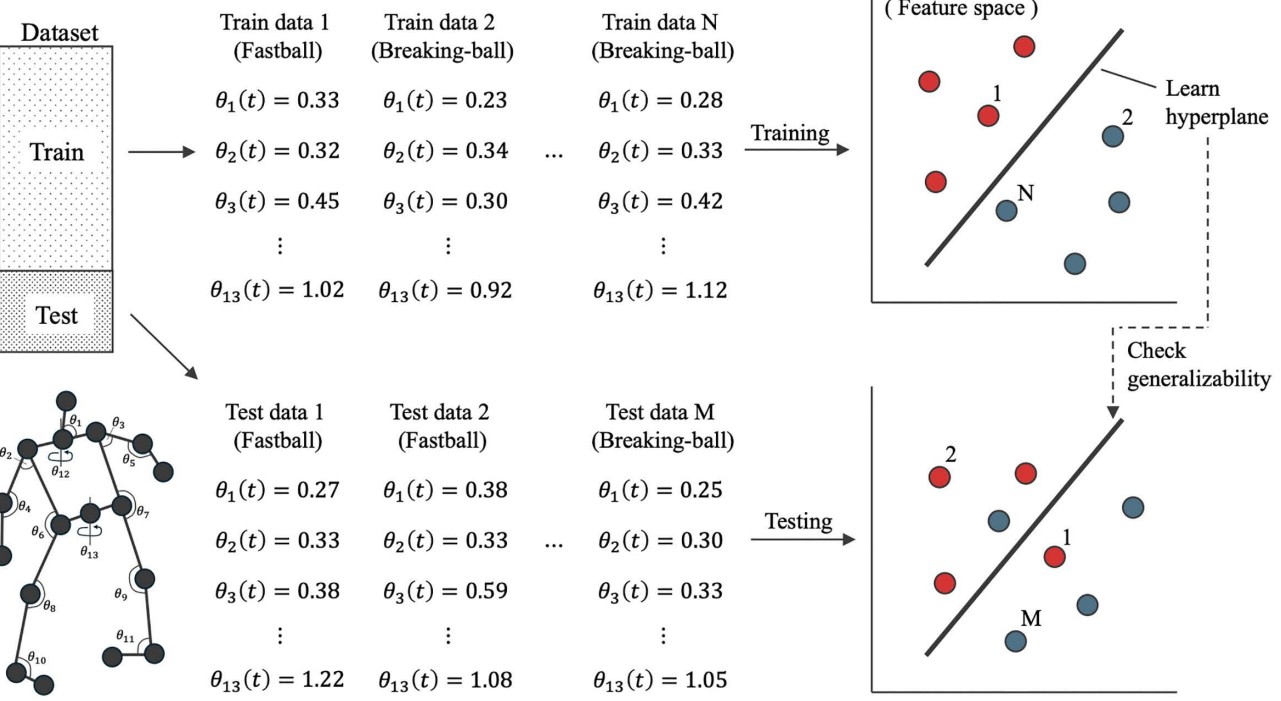

**Fig 2. Schematic image of ML-based pitch-type prediction using joint angles at time t as features.**

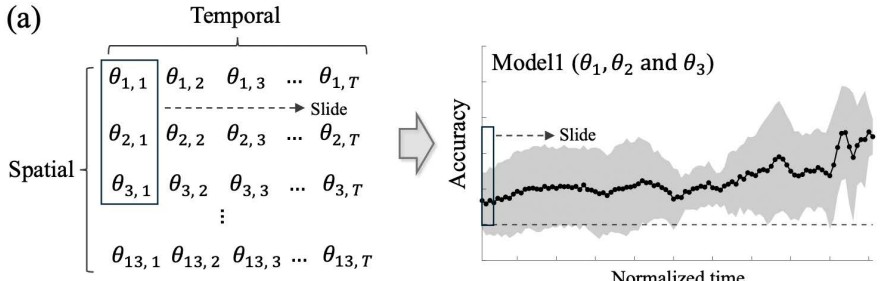

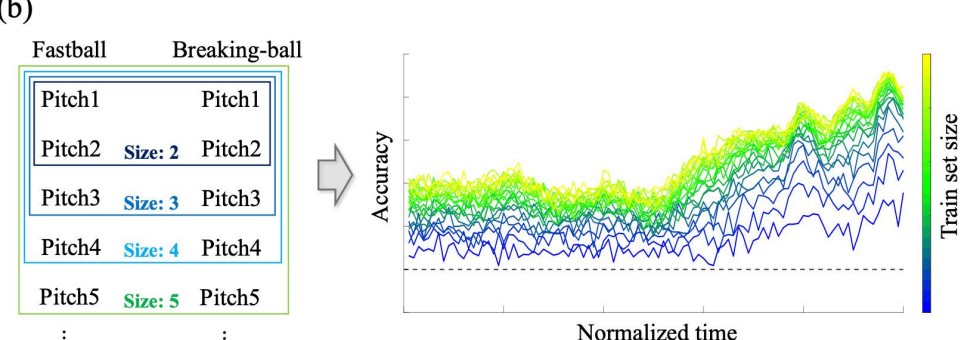

**Fig 3. Schematic illustration of the two analyses conducted in this study.** (a) Analysis 1: Sliding time-window analysis for identifying spatiotemporally informative cues for the ML model. (b) Analysis 2: Set-size analysis for evaluating how the accumulation of information across trials influences the prediction accuracy of the ML model.

Furthermore, in Analysis 2, we evaluated how the prediction accuracy changes over longer timescales as information about the target pitcher accumulated across trials, mirroring how batters incrementally learn about their opponents during real matches. This evaluation was achieved by varying the size of the training dataset (Fig 3b). Through random resampling of the training data and repeated training and testing with different dataset sizes, ML enables data-driven evaluation of the incremental contribution of each additional trial and its impact on the prediction accuracy.

Although the transferability of these analytical results to human athletes should be interpreted with caution, this novel analysis is expected to complement conventional hypothesis-testing experiments and serve as a foundation for follow-up studies that advance the understanding of predictive abilities.

## 2.2 Dataset collection and processing

**2.2.1 Information on the pitchers included in the dataset.** The dataset consists of data from eight college league pitchers ($174.1 \pm 4.1$ cm, $74.91 \pm 3.8$ kg, $19.87 \pm 1.0$ years). All pitchers had more than 10 years of playing experience. This study used previously collected, unpublished data. An opt-out consent framework was adopted in accordance with the university's ethical guidelines. Information about the study, including its purpose and data usage, was made publicly available to ensure transparency. Individuals whose data were included had a clear opportunity to decline participation. The Institutional Ethics Committee of the National Institute of Fitness and Sports in Kanoya approved (approval number: 25-1-26) the study procedure. All procedures adhered to the principles of the Declaration of Helsinki.

**2.2.2 Motion measurement and calculation of joint angles as the input feature.** For dataset collection, the motion of each pitcher was measured at a sampling rate of 200 Hz using 16 synchronized optical motion-capture cameras (Raptor-E and 16 Kestrel2200 cameras; Motion Analysis Corp, Santa Rosa, USA). Pitchers threw random pitch types at

random locations during the experiment to simulate real game-like conditions. The positions of 15 joints—the parietalis (head), both sides of the acromions (shoulders), lateral epicondyles of the humerus (elbows), radial styloid processes (wrists), greater trochanters of the femur (hips), lateral condyles of the femur (knees), and heels and tops of the shoes (toes)—were extracted from the measured data. Based on the measured joint-position data, time-series features were computed as follows: ten three-dimensional joint angles—both shoulders ($\theta_2$, $\theta_3$), elbows ($\theta_4$, $\theta_5$), hips $\theta_6$, $\theta_7$, knees $\theta_8$, $\theta_9$, and ankles ($\theta_{10}$, $\theta_{11}$)—as illustrated in Fig 2, and three two-dimensional angles (sagittal-plane angle between the head and shoulder vector and the horizontal axis of the ground ($\theta_1$) and the horizontal-plane rotation angle of the shoulder ($\theta_{12}$) and hip ($\theta_{13}$)). Data from the left-handed pitchers were mirrored to standardize the meaning of each angle.

The ball velocity (initial speed), ball crossing position over the home plate, and vertical and horizontal displacements of each pitch were also recorded as performance indices using the TrackMan system (TrackMan). Pitch type information was collected based on the participants' self-reports. A pitch was classified as a breaking ball if its average velocity was at least 10 km/h (6.21 mph) lower than the average fastball velocity of the pitcher. The breaking balls include change-ups, curveballs, sliders, sinkers, or splitters. To evaluate the extent to which the trajectory of each pitcher's breaking ball deviated from their fastball, the root mean squared value was calculated based on the differences in average vertical and horizontal displacements between the two pitch types. The minimum displacement between fastballs and breaking balls was at least 14 cm for all pitchers, which is larger than the approximate diameter of a baseball (~7 cm). Table 1 presents the summary of the fastballs and breaking-balls information for each pitcher.

Notably, because the pitchers simulated game-like situations by throwing random pitch types, the distribution of pitches across types was unbalanced. The potential effects of this imbalance, along with other individual differences, on the prediction accuracy of the ML models were also examined.

**2.2.3 Identification of the analysis phase and dataset development.** Subsequently, for each pitch, features from the onset of the pitcher's wind-up to the point of ball release were extracted and normalized to 101 time points. Wind-up onset was defined as the frame at which the vertical velocity of the leading knee dropped below 5% of its maximum value, traced backward from the moment when the knee reached its peak vertical position. The ball release was defined as the frame at which the resultant velocity of the throwing-arm wrist reached its maximum. Illustrations of the 20% intervals of the normalized motion are shown in Fig 4. Considering these preprocessing steps, a dataset consisting of 13 joint angles × 101-time frames × (number of pitches) was developed for each pitcher. The pitching motion data for each pitcher is available at GitHub (https://github.com/takamido/Pitch_type_pred_ML).

## 2.3 ML model design for the analysis

**2.3.1 Logistic regression model for pitch type prediction task.** Consistent with previous approaches used for EEG data analysis [20], logistic regression was employed as the classification model. Logistic regression is a linear method

**Table 1. Summary of the fastballs and breaking-balls information for each pitcher.**

| | Fastball pitch number | Breaking-ball pitch number | Mean fastball velocity (mph) | Mean breaking-ball velocity (mph) | Mean breaking-ball displacement (cm) |
|---|---|---|---|---|---|
| Sub1 | 56 | 62 | 69.44 | 62.83 | 16.23 |
| Sub2 | 57 | 48 | 76.09 | 63.74 | 24.01 |
| Sub3 | 37 | 48 | 76.41 | 64.65 | 14.10 |
| Sub4 | 55 | 59 | 73.83 | 66.16 | 17.42 |
| Sub5 | 56 | 44 | 75.79 | 67.01 | 26.59 |
| Sub6 | 47 | 50 | 80.18 | 65.55 | 37.47 |
| Sub7 | 30 | 40 | 75.41 | 63.40 | 28.21 |
| Sub8 | 80 | 31 | 76.17 | 63.86 | 32.93 |

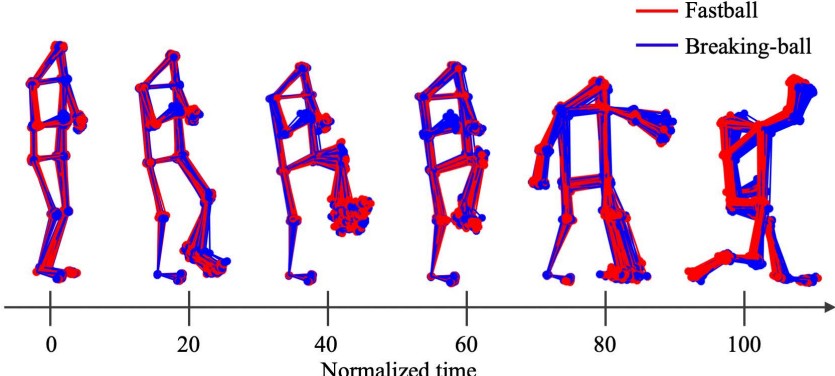

**Fig 4. Illustrations at 20% intervals of the normalized motion, using Sub1 as an example.**

that estimates the probability of class membership using a logistic function applied to the weighted sum of input features. Specifically, given the joint angles at time $t$ as input ($\Theta_t$), the logistic regression model calculated the probability that the pitch was a fastball using the following equation:

$$P_t(Y_t = 1|\Theta_t) = \sigma(z_t) = \frac{1}{1+e^{-z_t}},$$

(1)

where $z_t = \beta_0 + \beta_1\theta_{1,t} + \beta_2\theta_{2,t} + \ldots + \beta_{13}\theta_{13,t}$ represents the linear combination of input features, and $\sigma$ is the sigmoid function. When prediction is performed using only specific joint angles, for example $\theta_1$ and $\theta_2$, the expression reduced to $z'_t = \beta_0 + \beta_1\theta_{1,t} + \beta_2\theta_{2,t}$. Based on Equation (1), the model sought to determine a separating hyperplane between the fastball and breaking-ball data points up to a 13-dimensional joint-angle space (Fig 2).

**2.3.2 Justification of the model and input feature design.** From the perspective of understanding human behavior in a sports context, the ML analysis design of this study has the following advantages. First, the logistic regression is a representative explainable artificial intelligence model [22], in which the importance of each input feature is directly reflected in the magnitude of its corresponding weight ($\beta$). Furthermore, the postural information employed as an input feature in this study is a common predictive cue that has been identified in many previous studies with human athletes [23]. Although more complex features—such as joint angular velocity, angular acceleration, and their time-series information—can also be used as input features, increasing the complexity of the input information carries the risk of reducing interpretability and limiting its transferability to human athletes. Finally, this combination of simple models and features is considered well-suited for motion analysis in sports settings, where it is often difficult to collect large-scale datasets consisting of thousands to tens of thousands of samples.

Therefore, this study employed a relatively simple ML analysis design. However, the effects of modifying the ML design, such as incorporating additional features like angular velocity or using ML models capable of handling time-series data, such as LSTM (Long Short-Term Memory Network) [24], were also examined in the analysis section.

## 2.4 Analysis 1: Sliding time window analysis

**2.4.1 Model training and prediction performance evaluation.** In Analysis 1, we aimed to identify the spatiotemporal features relevant to pitch-type prediction of the ML model by applying a sliding time-window approach to the joint-angle time series. Specifically, for each pitcher, the logistic regression model was trained and tested using the feature vectors corresponding to a given time point $t$ within the 101 normalized time points. We continuously evaluated classification performance across the entire pitching motion by sliding the time window and repeatedly training and testing the model.

We further examine spatial importance by conducting independent evaluations under seven conditions: one using all 13 joint features simultaneously and six dividing the joints according to body regions (e.g., throwing arm: $\theta_2$ and $\theta_4$, leading arm: $\theta_3$ and $\theta_5$, trunk tilt: $\theta_1$, $\theta_6$ and $\theta_7$, trunk horizontal rotation: $\theta_{12}$ and $\theta_{13}$, pivot leg: $\theta_8$ and $\theta_{10}$, leading leg: $\theta_9$ and $\theta_{11}$). We repeated the following procedure 100 times for each time point $t$ to ensure a robust estimation, resulting in $7 \times 101 \times 100 = 70{,}700$ evaluations per pitcher:

(1) Data partitioning: For each pitch type, 25 trials (50 in total) were randomly sampled as training data, and five trials (10 in total) were sampled as test data. This balanced sampling procedure corrected data imbalances across pitchers, maintained a consistent chance level, and ensured that the classifier was trained with equal numbers of fastball and breaking-ball trials while keeping the test sets independent. Similar to bootstrap methods [25], this approach reduced the sensitivity to extreme data points by repeatedly utilizing subsets of the overall dataset, thereby stabilizing the mean values used in subsequent statistical analyses.

(2) Preprocessing: Input features were standardized to a standard normal distribution based on the training set. Normalization parameters (mean and variance) estimated from the training data were then applied to normalize the test data. This procedure prevented information leakage, which would otherwise have occurred if the training and test data were normalized together.

(3) Model training and evaluation: A logistic regression model with an iterative solver (maximum 1,000 iterations) was fitted to the training set. The *lbfgs* solver implemented in the *sklearn.linear_model* library in Python was employed with L2 regularization. After fitting, classification was performed on the test data, and the prediction accuracy of the independent test set was recorded.

**2.4.2 Statistical analysis of the prediction results.** As a result of the above analysis, the average accuracies at 101 time points were obtained for each of the eight pitchers under the seven input feature conditions. Accordingly, we performed statistical analyses to determine which spatiotemporal features contributed significantly to the discriminative performance of the ML model.

First, to examine whether the logistic regression model could predict pitch type from full joint-angle information at a level significantly above chance (0.50), we applied a cluster-based permutation test [26] using the mean and standard deviation across the eight pitchers under the all-joint condition. Cluster-based permutation testing was conducted using 10,000 permutations. The cluster-forming threshold was set at $p < 0.05$ (one-sided), and the cluster-level significance criterion was set at $\alpha = 0.05$. Cluster mass was defined as the sum of $t$-values across contiguous time points above the threshold, and significance was determined relative to the permutation-based null distribution generated through random sign flips. To verify the effect size of the identified cluster, the maximum effect size within the cluster ($d_{max}$) and the effect size averaged over the rectangular time window circumscribing the cluster were calculated ($d_{rect}$) [27]. As discussed previously [27], $d_{rect}$ tends to yield smaller values than $d_{max}$, serving as a more conservative estimate or safety margin. However, in some cases, it may result in larger values depending on the data distribution. Clusters shorter than ten consecutive time points (corresponding to approximately 0.2 s) were excluded from the analysis.

Furthermore, for each of the six individual body-region conditions, a cluster-based permutation test was applied to the averaged data across the eight pitchers using the same procedure. The initial cluster-forming significance criterion was set at $\alpha = 0.05$. We accounted for multiple comparisons by adjusting the significance levels of the identified clusters using the Holm–Bonferroni method across the six body regions. The effect size ($d_{max}$ and $d_{rect}$) was also calculated.

**2.4.3 Additional analyses to evaluate factors affecting ML prediction.** In addition to the main analysis described above, we conducted additional analyses to examine the factors influencing the predictive performance of the ML model. Although improving the predictive performance of ML models is not the primary aim of this study, they provide insights into potential factors that affect model performance in the ML-based analysis of athletic movements, which typically rely

on small-scale datasets and exhibit substantial individual variability. Specifically, the effects of the following factors were evaluated.

(1) Angle velocity and acceleration information: Although joint angle information was utilized in the analysis, incorporating additional features such as angular velocity and acceleration may further improve the performance of the ML model. Therefore, we conducted the same training and evaluation procedures using datasets that included the angular velocity and angular acceleration of each joint as additional input features (resulting in input dimensions of 26 and 39, respectively) to assess their impact. Considering the prediction accuracies of the models trained on these datasets, comparisons with the prediction accuracy obtained using the original joint angle datasets were conducted using cluster-based permutation tests.

(2) Usability of time-series information. Although the logistic regression model used in this study can only account for postural information at individual time points, incorporating temporal dependencies, that is, utilizing time-series data, may enhance the model's performance. Therefore, representative ML models for time-series data processing, specifically, the Gated Recurrent Unit (GRU) [28] and LSTM networks [24], were also implemented to evaluate their prediction performance. A single hidden layer with 64 units was used for each model, and training was performed for 80 epochs. The input at each time step t consisted of cumulative postural information from the initial frame up to time $t$ (i.e., 13 joints × $t$ frames). The prediction accuracies of these models were compared with those of the logistic regression model using cluster-based permutation tests.

(3) Differences in the conditions across datasets. Finally, this study verified the effects of the differences in the conditions across individual pitchers' datasets. Specifically, we calculated the correlation coefficient between the mean prediction accuracy across the entire time frame for each pitcher and the following five parameters. Each of the computed correlation coefficients was evaluated for statistical significance using a test of zero correlation (null hypothesis of no correlation).

(a) Variance of pitch location: This metric indicates the degree of variability in the ball's crossing position at home plate of each pitcher. It is defined as the root sum square of the standard deviations in the vertical and horizontal (measured from the right to the left batter's box) directions, calculated across all pitch types.

(b) Velocity difference between fastballs and breaking balls: Indicates the magnitude mean speed difference between the fastball and breaking ball of each pitcher.

(c) Breaking ball displacement: Indicates the magnitude of the movement of the breaking balls relative to the fastballs. It is defined as the root sum square of the vertical and horizontal displacements of the ball.

(d) Pitching motion variability: Represents the degree of variation in the pitching motion across the trials. It is defined as the average joint angle variability over time, where the variability at each time frame is calculated as the sum of the standard deviations of the joint angles across trials.

(e) Pitch type imbalance: Indicates the imbalance in the number of fastballs and breaking balls thrown. It is defined as the ratio of the larger to the smaller number of pitches between the two pitch types.

(f) Total pitches: The total number of pitches thrown by each pitcher, including fastballs and breaking balls.

## 2.5 Analysis 2: Set size analysis

In Analysis 2, we evaluated how the prediction accuracy of the ML model changed over longer timescales as information about the target athlete accumulated across the trials. Specifically, for each pitcher, the training dataset size was

incrementally increased from 2 to 25 trials per pitch type (i.e., 4–50 trials in total), and the logistic regression model was repeatedly trained and tested. The test dataset was fixed at five trials per pitch type (ten trials in total), independent of the training dataset size. For each dataset size, the training and test data were randomly shuffled, and the accuracy was evaluated 100 times at each of the 101 time points. Data preprocessing and logistic regression model fitting were performed using the same procedures as in Analysis 1.

From the above analysis, the mean prediction accuracies were obtained for each pitcher, and the dataset size was calculated at each of the 101 time points. Subsequently, we computed the mean accuracies for dataset sizes of 2, 5, 10, 15, 20, and 25 trials per pitch type. Differences in the mean accuracy across the five intervals (2–5, 5–10, 10–15, 15–20, and 20–25 trials) were then calculated, and the means and standard deviations were obtained across individuals. We reduced the impact of multiple comparisons and mitigated noise by making comparisons over larger spans rather than computing differences at single-trial increments. We verified whether the increases in accuracy across each interval were significantly greater than the chance level (0.0) by applying cluster-based permutation tests to all five intervals. These tests were conducted following the same procedure as in Analysis 1, and significance levels were adjusted for multiple comparisons using the Holm–Bonferroni method.

## 3 Results

### 3.1 Analysis 1: Sliding time window analysis

**3.1.1 Model performance.** Fig 5 illustrates the mean prediction accuracy and standard deviation under the all-joint feature condition at each time point for each of the eight pitchers. Fig 6 summarizes the overall mean and standard deviation across all pitchers. As shown in the figures, although individual differences existed in overall accuracy and temporal patterns, the ML model demonstrated prediction accuracy above the chance level across most time points when all joint features were used, showing a general trend of increasing accuracy as the ball release approached. The cluster-based permutation test of mean accuracy across pitchers under the all-joint condition revealed that the entire interval constituted a single significant cluster ($p < .01$; $d_{max} = 8.26$, $d_{rect} = 2.92$).

Fig 7 and 8 shows the mean prediction accuracy for each of the six body regions, with the results of all eight pitchers overlaid. The cluster-based permutation test for six regions revealed that classification based on the throwing-arm (8–100%), leading arm (60–100%), trunk tilt (0–58% and 65–100%), trunk horizontal rotation (66–100%), pivot leg (65–100%) and leading leg (70–96%) achieved prediction accuracies significantly above the chance level (*adjusted* $p = 0.019, < .01, 0.016, 0.013, < .01, < .01$ and $< .01$, $d_{max} = 4.94, 2.86, 2.26, 3.92, 2.89, 2.13$ and $1.96$, $d_{rect} = 1.96, 2.76, 1.77, 3.21, 2.11, 2.13$ and $1.66$, respectively). In terms of effect size, $d_{max}$ was largest for the throwing arm, while $d_{rect}$ was highest in the later phase of trunk tilt.

As shown in the individual plots (Figs 5 and 7), there were individual differences in terms of the body regions used at different time points and the extent to which such information enabled prediction. For example, while participant 2 achieved an accuracy of approximately 80% using information from the right arm in the early phase, participant 7 demonstrated high accuracy based on information from the trunk in the latter phase (Fig 8).

**3.1.2 Factors affecting ML prediction performance.** Fig 9 and Table 2 present the results of this additional analysis. Fig 9 illustrates the effects of varying the ML model and the input features on prediction accuracy. The cluster-based permutation test results show no substantial differences attributable to model type or input features were observed ($p > 0.05$) at a statistically detectable level given the current sample size. However, datasets that included velocity and acceleration showed an average improvement of approximately 3–5% in the later phase of the sequence (after 80%) (Fig 9b).

Finally, Table 2 presents the correlation coefficients between various parameters (e.g., total pitches) and the prediction accuracies for each pitcher. According to the results of the tests for zero correlation, no parameters showed a significant correlation. Among the parameters examined in the additional analyses, pitch type imbalance and total number of pitches

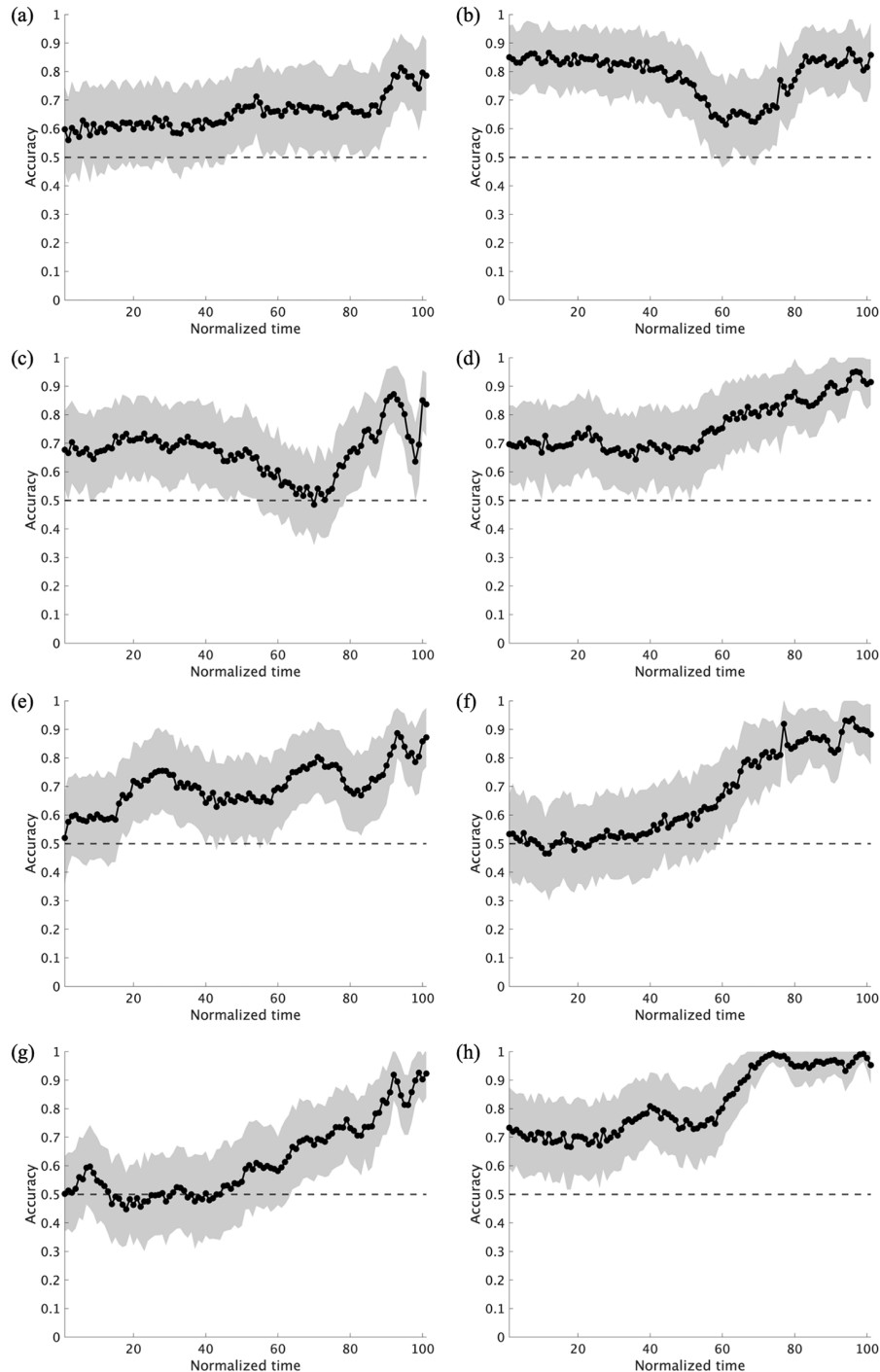

**Fig 5. Mean prediction accuracy and standard deviation for each pitcher at each time point using full-joint information.** Panels (a)–(h) show the individual plots for subjects 1–8.

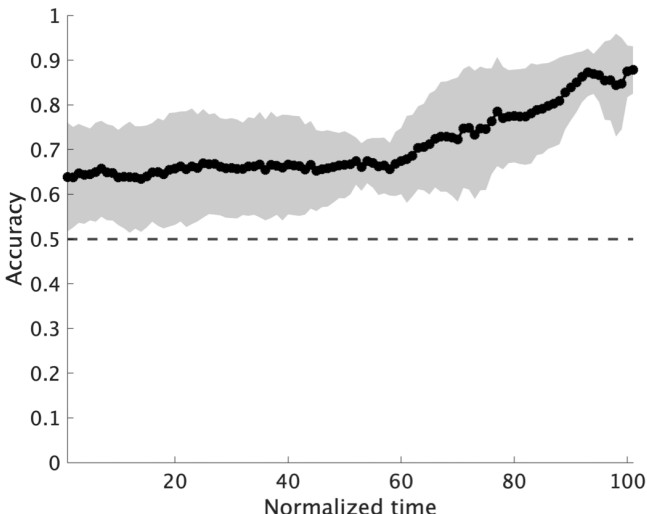

**Fig 6. Overall mean prediction accuracy and standard deviation across the eight pitchers using full-joint information.**

exhibited moderate correlations with prediction accuracy ($r > 0.50$), whereas breaking ball characteristics—such as displacement and velocity difference from fastballs—showed only weak correlations ($r < 0.10$).

### 3.2 Analysis 2: Set size analysis

Fig 10 shows the relationship between dataset size and prediction accuracy at each time point for each pitcher. According to the cluster-based permutation test results, significant improvements were observed in the following intervals: for training dataset sizes of 2–5 trials at 13–36% and 38–100% (*adjusted p* = 0.016 and <.01, $d_{max}$ = 3.29 and 6.73, $d_{rect}$ = 1.59 and 2.45, respectively); for 5–10 trials at 3–17% and 19–100% (*adjusted p* = 0.018 and <.01, $d_{max}$ = 2.38 and 4.39, $d_{rect}$ = 1.47 and 3.24, respectively); for 10–15 trials at 30–40%, 50–73%, and 75–100% (*adjusted p* = 0.020, <.01 and <.01, $d_{max}$ = 1.49, 2.63, 2.46, $d_{rect}$ = 1.25, 3.02, 3.35, respectively); and for 15–20 trials at 86–98% (*adjusted p* < .01, $d_{max}$ = 2.92, $d_{rect}$ = 2.51) (Fig 11). No significant clusters were observed in the trial interval of 20–25. These results demonstrate that the impact of adding additional training data on prediction accuracy is more pronounced when the dataset size is small.

### 4 Discussion

#### 4.1 Similarities and differences between the human athletes and ML model

The findings of Analysis 1 revealed several notable parallels with those previously reported in experts' anticipation skills. (1) In line with previous findings that distal segments directly linked to ball release provide key anticipatory cues [29,30], the most informative cues for the ML model were derived from the throwing arm. (2) Consistent with reports that proximal information can also support anticipation [7,31], the ML model achieved above-chance accuracy even when using only proximal cues, such as those from the trunk. (3) As suggested by studies emphasizing the integration of multiple information sources [7,8,31]; the highest prediction accuracy was observed when proximal and distal information was available.

Therefore, the cues effective for human athletes are also embedded within those utilized by the ML model. The improvement in prediction accuracy achieved by combining distal and proximal cues suggests a complementary relationship between them; that is, they provide qualitatively distinct types of information. Therefore, the cognitive strategy commonly reported in athletic anticipation skills may be valid not only behaviorally, but also from a purely informational and computational standpoint.

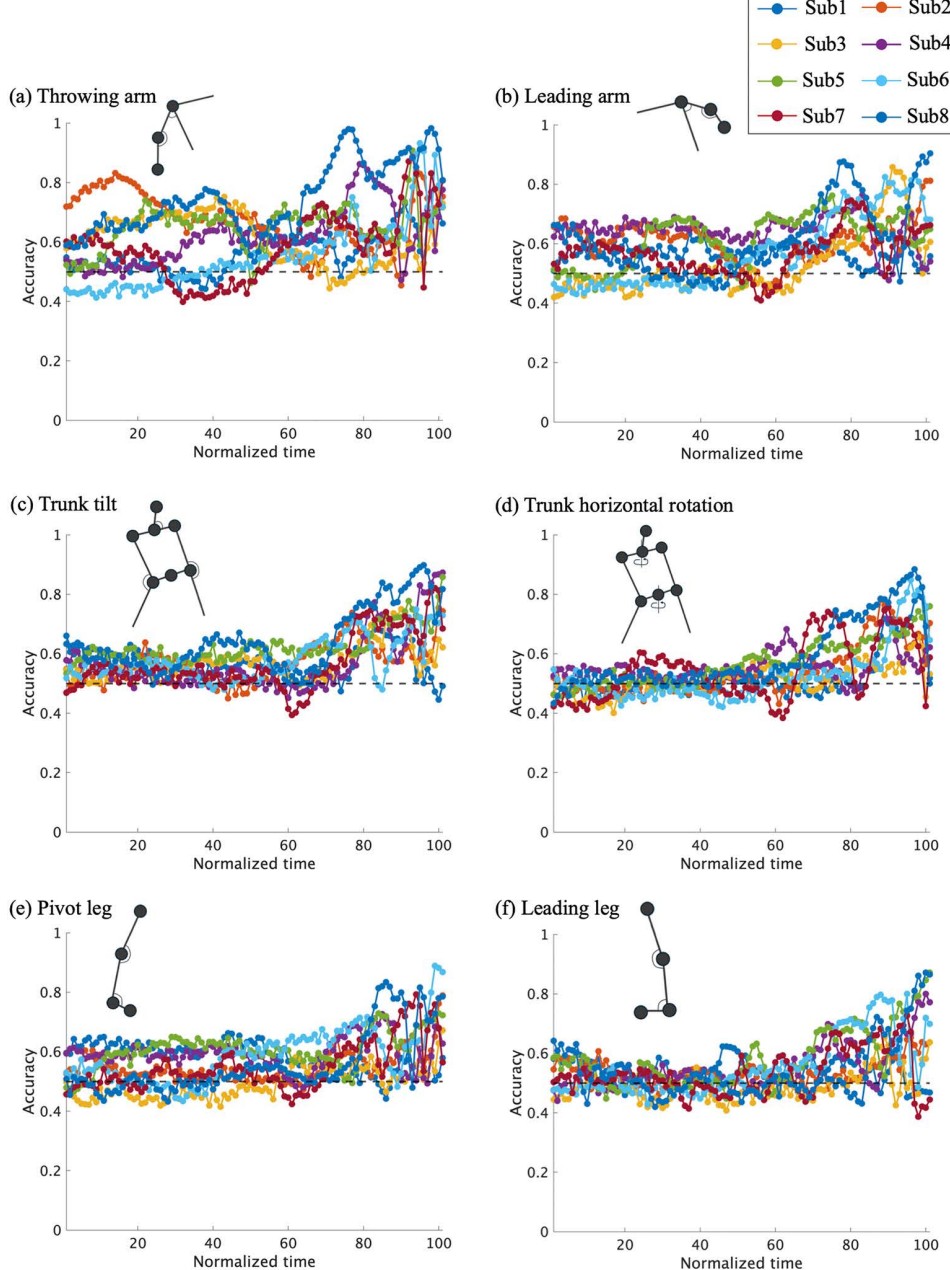

**Fig 7. Mean prediction accuracy for each of the six body regions, with results from all eight pitchers overlaid.**

However, Analysis 1 also revealed several discrepancies compared to previous findings on human athletes. (1) High prediction accuracy exceeding 70% was achieved even when using only distal body parts that do not directly influence the ball trajectory, such as the pivot leg. (2) Although many prior studies involving human athletes have reported above-chance prediction accuracy around ball release [6,32], the ML model in this study was able to make accurate predictions from the very beginning of the wind-up phase. (3) Dynamic features, such as joint velocity and acceleration, and the temporal accumulation of information, had little effect on the prediction performance of the model.

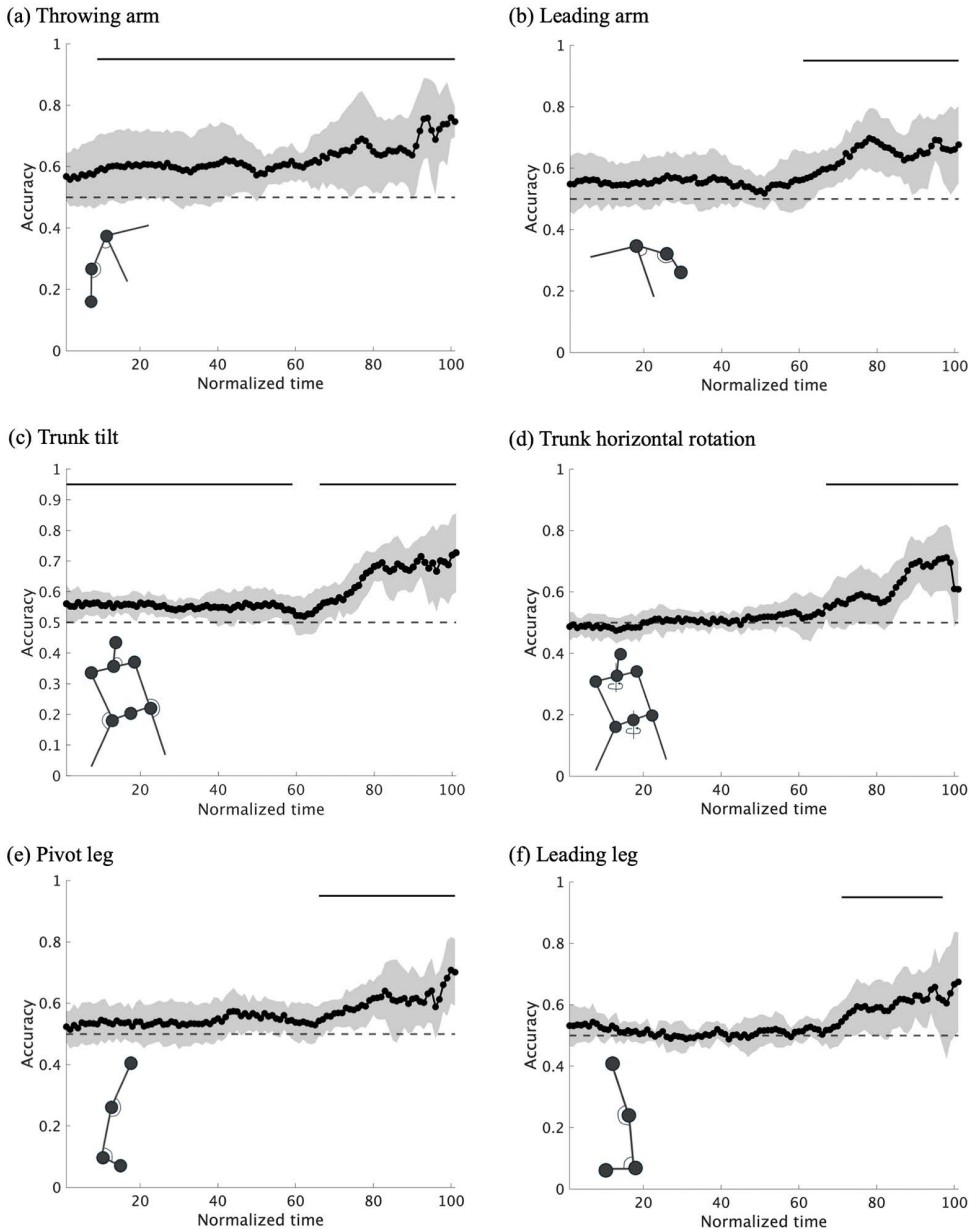

**Fig 8. Mean prediction accuracy and standard deviation across eight pitchers for each of the six body regions.** Solid lines indicate significant intervals (p < .05).

These findings suggest that, compared with human athletes, the ML model can make accurate predictions based on more spatially and temporally limited cues. One possible explanation for this is that, while humans tend to integrate multiple sources of information over time to mitigate perceptual noise [33], the ML model was unaffected by such noise and had access to complete and accurate information at each time point. Therefore, although (1) and (2) may represent overlooked cues that are also available to human athletes, the athletes may prioritize cues that are more robust against perceptual noise.

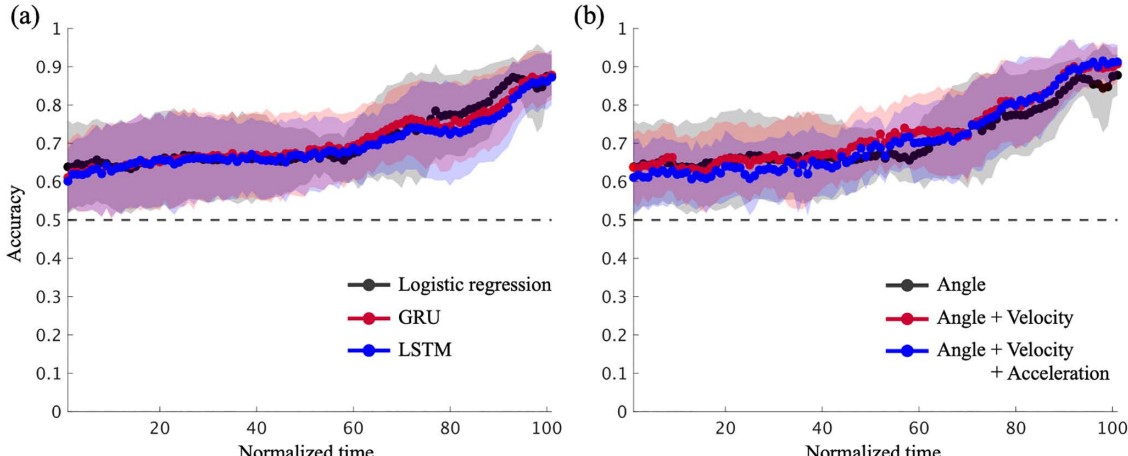

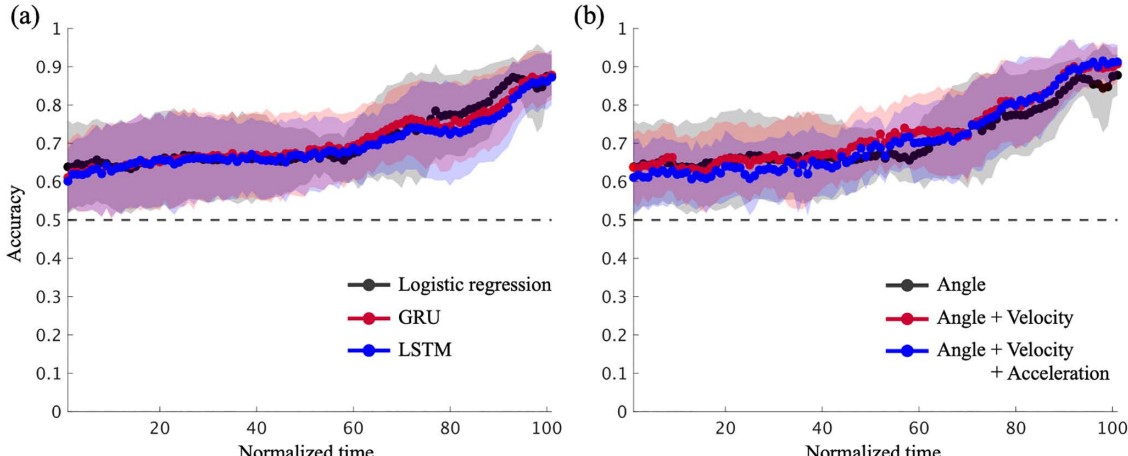

**Fig 9. (a) Comparison of prediction accuracy among logistic regression, GRU, and LSTM models.** (b) Comparison of prediction accuracy of models trained using (i) joint angle information, (ii) joint angles and angular velocity, and (iii) joint angles, angular velocity, and angular acceleration.

**Table 2. Relationship between each parameter and the prediction accuracy of each pitcher.**

| Parameter | Correlation coefficient with prediction accuracy |
|---|---|
| Variance of pitch location | −0.439 |
| Velocity difference between fastballs and breaking balls | 0.025 |
| Breaking ball displacement | 0.093 |
| Pitching motion variability | −0.179 |
| Pitch type imbalance | 0.512 |
| Total pitches | 0.597 |

## 4.2 Individual differences in the predictive cues and prediction accuracy

Furthermore, the individual differences in predictive cues observed in Analysis 1 highlight the value of the analysis as a complementary approach to conventional experimental studies. In conventional experimental studies, visual stimuli are typically derived from the movements of a few players; hence, the differences in the cues identified in these studies may be attributable to individual variability. Therefore, it may be ideal to conduct a pre-ML analysis to clarify the spatiotemporal characteristics of the cues embedded in target players before conducting a controlled experiment. As discussed in the Introduction, such individual variability in cues has been difficult to assess using conventional experiments involving real human participants. Thus, the ability of the proposed ML analysis to explore individual-specific predictive cues in a data-driven manner may be the key strength of the present approach.

However, this study did not identify the specific factors responsible for individual differences in the ML prediction accuracy. According to the results of the additional analysis, pitchers who generated larger deviations from fastball trajectories were not necessarily easier for the ML model to predict, and the magnitude of trial-to-trial variability in each pitcher's motion also failed to explain the differences in prediction accuracy across individuals. Therefore, to understand the individual differences in prediction accuracy observed in this study, it may be necessary to conduct more in-depth, individual-specific analyses of pitching motions, such as how each pitcher's individual coordination patterns [34,35] change between fastballs and breaking balls, rather than searching for common explanatory factors across individuals. Because dataset

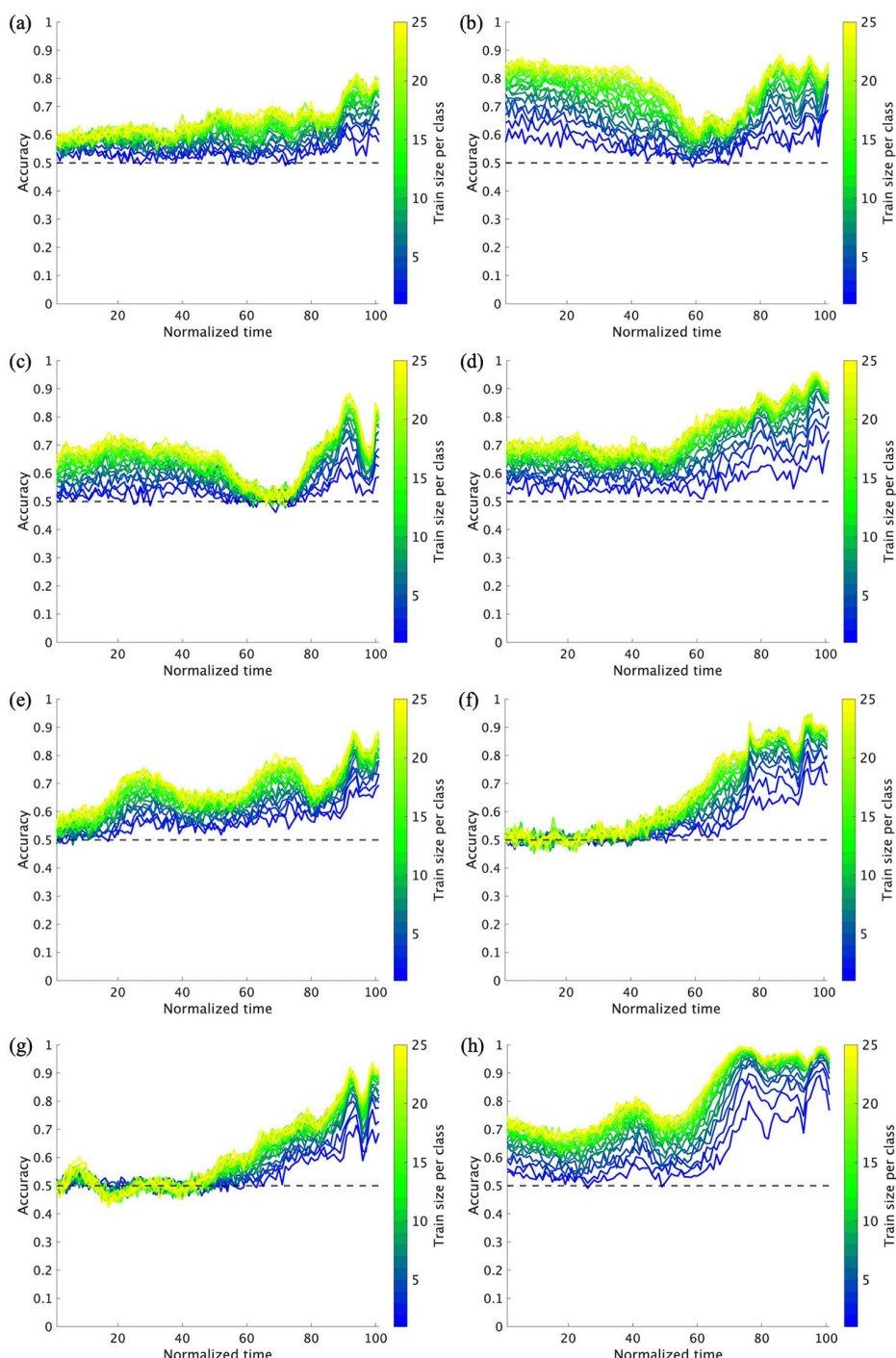

**Fig 10. Relationship between dataset size and prediction accuracy at each time point for each pitcher.** Panels (a)–(h) show individual plots for subjects 1–8.

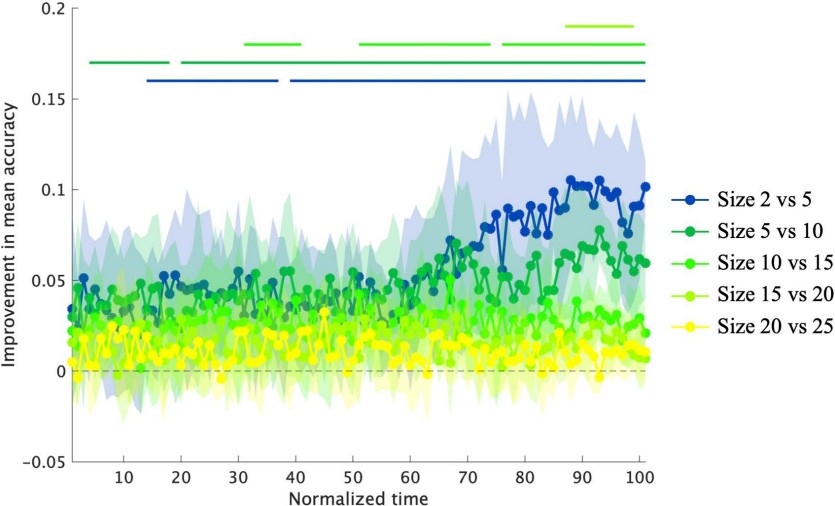

**Fig 11. Improvement in mean prediction accuracy with increasing training data.** Numbers on the dataset size axis represent the number of trials per pitch type (thus, a size of 5 corresponds to 10 training trials in total). Solid lines indicate intervals where accuracy improvements were statistically significant compared with the chance level (0.0).

size and class imbalance showed moderate correlations with prediction accuracy ($r > 0.50$), future studies should control for these factors across individuals to enable more detailed investigation into the sources of individual differences.

## 4.3 Information accumulation effect on the prediction accuracy

Notable improvements in prediction accuracy were observed in the analysis with dataset sizes of up to approximately 15 pitches per condition (i.e., 30 in total), suggesting that each individual pitch carries relatively more informative value in the early stages when the cumulative number of pitches is low. The greater sensitivity to dataset size in the latter part of the motion is likely due to the increased movement variability near the ball release, with each additional data point contributing more substantially to reducing the uncertainty.

From a real game perspective, the improvement in accuracy resulting from the accumulation of pitch information may be associated with the Times Through the Order Penalty, which refers to the phenomenon in which batters tend to improve their performance when facing the same pitcher multiple times. Although the underlying causes of this phenomenon remain debatable [36], the present findings suggest that a reduction in uncertainty and an increase in predictive accuracy resulting from the accumulation of information may be contributing factors. However, these findings should be regarded as computational analogies rather than direct evidence of cognitive mechanisms. Therefore, further controlled experiments are required to verify the hypotheses generated from these results.

## 4.4 Practical implications and future directions

The findings of the analyses yielded several practical insights. First, for pitchers, ML may be leveraged to train and adopt movements that reduce their predictability, thereby facilitating the development of deceptive or disguised motions [37]. Second, information on pitcher-specific predictive cues could support batters by enhancing their anticipatory strategies. Finally, evaluating individual differences in predictability could contribute to scouting and talent identification from a novel perspective, namely, the degree to which a player is difficult to predict by opponents. However, it is important to note that these practical implications should be considered only after further empirical validation, specifically to confirm whether the cues identified by the ML model are indeed accessible and usable by human observers.

Finally, this study has some limitations and offers directions for future work. First, whether the cues of individual pitchers identified through the ML analysis in this study are accessible and usable by human athletes remains to be carefully examined in future research. Moreover, because this study focused on intrapersonal training and testing, future studies should examine the generalizability of spatiotemporal cues across individuals. From this perspective, because this study employed a relatively small sample ($n=8$) consisting of university-level athletes, the generalizability of the findings is limited, and constructing a larger dataset that includes a more diverse and higher-caliber athlete sample is necessary in future studies. Finally, as this study focused on pitch type prediction in baseball, future research should examine the applicability of the proposed analysis to a broader range of predictive tasks across sports.

## 5 Conclusion

Data-driven analysis using ML yielded insights that can inform follow-up experiments and deepen the interpretation of existing findings. These include potential cues that enhance the model's prediction accuracy, individual differences in cue informativeness, and the effect of information accumulation across trials. Although alignment with human athletes should be carefully examined in future studies, a key theoretical contribution of this study is that it offers a time-resolved, data-driven account of where and when pitch-type–predictive information emerges in pitching kinematics. By addressing potential limitations and outlining directions for future research, the proposed ML analysis is expected to complement hypothesis-driven anticipation studies by enabling the quantification of predictive information without the need to prespecify candidate cues.

## Author contributions

**Conceptualization:** Ryota Takamido, Hiroki Nakamoto.

**Data curation:** Chiharu Suzuki.

**Formal analysis:** Ryota Takamido.

**Funding acquisition:** Ryota Takamido.

**Investigation:** Ryota Takamido, Chiharu Suzuki.

**Methodology:** Ryota Takamido, Hiroki Nakamoto.

**Project administration:** Chiharu Suzuki.

**Resources:** Hiroki Nakamoto.

**Supervision:** Hiroki Nakamoto.

**Validation:** Chiharu Suzuki, Hiroki Nakamoto.

**Visualization:** Ryota Takamido.

**Writing – original draft:** Ryota Takamido.

**Writing – review & editing:** Chiharu Suzuki, Hiroki Nakamoto.

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
