## [Decision Letter · Decision Letter 0]

18 Dec 2025

PONE-D-25-57679A data-driven analysis of spatiotemporal cues and experience accumulation effects for pitch type predictionPLOS One

Dear Dr. Takamido,

Thank you for submitting your manuscript to PLOS ONE. After careful consideration, we feel that it has merit but does not fully meet PLOS ONE’s publication criteria as it currently stands. Therefore, we invite you to submit a revised version of the manuscript that addresses the points raised during the review process.

We look forward to receiving your revised manuscript.

Kind regards,

Esedullah Akaras

Academic Editor

PLOS One

[This work was supported by the Japan Society for the Promotion of Science (Grant number: JP25K21018).].

Additional Editor Comments (if provided):

Reviewers' comments:

Reviewer's Responses to Questions

**Comments to the Author**

1. Is the manuscript technically sound, and do the data support the conclusions?

Reviewer #1: Yes

Reviewer #2: Yes

2. Has the statistical analysis been performed appropriately and rigorously? 

Reviewer #1: Yes

Reviewer #2: Yes

3. Have the authors made all data underlying the findings in their manuscript fully available?

Reviewer #1: Yes

Reviewer #2: Yes

4. Is the manuscript presented in an intelligible fashion and written in standard English?

Reviewer #1: Yes

Reviewer #2: Yes

5. Review Comments to the Author

Reviewer #1: This study endeavors to identify the spatio - temporal cues for predicting baseball pitch types and analyze the experience accumulation effect by means of data - driven machine learning approaches. The research perspective holds a certain degree of novelty, and the methodological design is relatively systematic. Nevertheless:

### 1. Research Sample and Data

- **Limited Sample Size**: The sample consists of only 8 pitchers, all of whom are at the college level. This significantly restricts the generalizability of the research findings. It is advisable to explicitly state this limitation within the discussion section and propose that future research incorporate a more diverse and higher - caliber athlete sample.

- **Insufficient Basis for Pitch Type Classification**: The classification of pitch types is solely predicated on speed differentials (≥10 km/h) and pitchers' self - reports. The absence of objective pitch trajectory data (such as lateral or vertical displacements provided by TrackMan) may introduce classification biases.

- **Unbalanced Data**: There is an imbalance in the number of fastballs and off - speed pitches among pitchers. Despite attempts to balance the data during sampling, this may still impact the stability and generalization ability of the model during training.

### 2. Methodological Modeling and Interpretation

- **Unverified Association with Human Cognition**: The study highlights the adoption of a logistic regression model due to its interpretability. However, it fails to verify whether the "cues" identified by the model are actually utilized by human batters. It is recommended to clearly demarcate between "machine - identifiable information" and "information practically employed by humans" in the discussion.

- **Assumption of Temporal Independence**: The model is trained independently at each time point, overlooking the temporal dependence inherent in the action sequence. It is suggested to supplement the analysis with sequence models such as LSTM or GRU for comparative purposes and explore the influence of temporal information on prediction.

- **Oversimplified Feature Engineering**: Feature engineering is overly simplistic, relying solely on joint angles while neglecting dynamic features such as joint velocities and accelerations, which may contain more discriminative information.

### 3. Result Analysis and Interpretation

- **Over - Reliance on Cluster Permutation Tests in Statistical Analysis**: Although cluster permutation tests are suitable for time - series data, the failure to report effect sizes (such as Cohen's d) renders it arduous to evaluate the practical significance of the results.

- **Inadequate Analysis of Individual Differences**: While individual differences are acknowledged, a deeper analysis of their origins (such as pitching motion styles and pitch type combinations) is lacking. There is a dearth of in - depth exploration regarding why certain pitchers are more predictable.

- **Prudent Interpretation of the "Experience Accumulation Effect"**: The improvement in model accuracy with an increase in training samples does not necessarily equate to the "experience accumulation" of human batters. This should be clearly presented as a computational analogy rather than direct evidence of a cognitive mechanism in the discussion.

### 4. Manuscript Writing and Structure

- **Refinement of English Expression**: Some sentences are overly long and exhibit unnatural grammar. It is recommended to seek professional English editing services or the assistance of native speakers for language refinement.

- **Chaos in Figure Referencing**: The manuscript frequently refers to "Fig 5–Fig 10", yet some figures in the submitted file are labeled as "In review" or are missing, which severely hinders the review process.

- **Streamlining of the Discussion Section**: Some parts of the discussion section repetitively describe the results. It is advisable to strengthen the comparison with prior research and accentuate the theoretical contributions and practical implications of this study.

Reviewer #2: The manuscript aims to explore, through an innovative perspective, how data-driven analytics and machine learning can reveal spatiotemporal cues relevant to baseball pitch prediction. The study is distinguished by its complementary approach to conventional research, overcoming the limitations of hypothesis-based paradigms and allowing the identification of predictive information sources that, traditionally, might escape observation. The authors use ML models applied to time series of joint angles collected through motion capture, performing two distinct analyses—one on the temporal evolution of cues and the other on the effects of accumulating opponent-specific information.

The originality of the project lies in the integration of ML predictions with detailed analysis of body movement, providing new insight into the moments and regions where relevant cues appear, as well as how progressive experience with the same pitcher can influence predictive accuracy. The study thus makes a significant contribution to the understanding of the mechanisms of sports prediction, opening promising directions for future research and for combining computational methods with the assessment of real-life athlete behavior.

The introduction of the article is notable for its solid structure and convincing argumentation of the need for the study. The theoretical context of sports prediction is clearly presented, and the specialized literature is coherently integrated, which gives the text rigor and academic relevance. Also, the contributions made by the proposed method are well outlined: the identification of spatiotemporal cues, the use of real athlete data, and the analysis of the accumulation of opponent-specific information. The objectives are explicitly stated, and the originality of the ML-based approach is evident, which reinforces the innovative nature of the research.

However, the introduction could be improved by more concisely formulating the limits of conventional methods and by more directly clarifying the problems they raise, such as the dependence on pre-established hypotheses and the risk of omitting relevant cues. In addition, mentioning the methodological challenges related to the relatively small data set or individual variations would strengthen the justification for choosing the proposed method. A slightly more compact structuring of the section would facilitate reading and highlight more clearly the exact direction of the study's contribution.

The Materials and Methods section provides a rigorous and well-founded presentation of how the machine learning-based analysis for baseball pitch prediction was built. The choice of task is well-argued by its relevance to real-world game situations, and the use of individualized models for each pitcher faithfully reflects how athletes construct their knowledge of their opponents. At the same time, the integration of theoretical concepts — including the adaptation of modern neuroscientific frameworks — gives depth to the approach, and the clear presentation of the two analyses, together with the availability of the code on GitHub, supports transparency and replicability.

On the other hand, including a lot of technical information in a compact space can make the material difficult to navigate. Some methodological explanations, descriptions of the ML model, and theoretical justifications appear grouped in extended paragraphs, which the reader may require additional information. A more fragmented structuring and more direct restatement of certain concepts—for example, the reason why postural cues were treated independently at each temporal point—could make the presentation more accessible.

In the Discussion section, the authors manage to coherently capitalize on the results obtained, emphasizing both the relevance of spatial and temporal cues and the way in which they align with previous research. It is the merit of the text that it highlights original contributions, such as the emergence of predictive cues in earlier phases of the movement or the influence of information accumulation on anticipatory accuracy. The connection with applied sports phenomena, such as the “Times Through the Order Penalty”, strengthens the practical value of the conclusions. The analysis of differences between pitchers and the suggestion of future directions — including direct comparisons with athlete behavior or the expansion of the database — pertinently complete the interpretation of the results.

However, the exposition of ideas in the Discussions is sometimes very rich, which can make the thread of explanations harder to follow. Some passages include several interpretive directions at the same time, and certain details about the models or the structure of the temporal data would have been more appropriate in the methodological part. At the same time, the practical applicability could be further delimited by a clarification of the inherent limits of ML models in relation to the real perceptual strategies of athletes. A more visible thematic segmentation would facilitate the understanding of the evolution of the arguments.

6. PLOS authors have the option to publish the peer review history of their article (what does this mean?). If published, this will include your full peer review and any attached files.

Reviewer #1: No

Reviewer #2: **Yes:**Ilie Eva

---

## [Author Response · Author response to Decision Letter 1]

26 Jan 2026

Dear Editor and Reviewers,

We sincerely appreciate the valuable feedback and constructive comments provided by the reviewers and the editorial team. In response to their suggestions, we have significantly revised our manuscript. The major revisions are summarized as follows.

1. Additional analysis to evaluate potential factors affecting ML prediction accuracy

Based on the comment by Reviewer 1, we added an additional analysis to examine which factors influence the predictive performance of the ML model. Specifically, we investigated the impact of several factors, including (1) the use of time-series models such as GRU and LSTM, (2) the inclusion of dynamic features such as joint angular velocity and acceleration, and (3) differences across individual pitchers’ datasets, such as the degree of class imbalance. Although improving the predictive performance of ML models was not the primary aim of this study, we believe that it provides insights into the potential factors that affect model performance in the ML-based analysis of athletic movements, which typically rely on small-scale datasets and exhibit substantial individual variability.

2. Redesign of the overall manuscript structure

Based on the suggestions provided by both reviewers, we made substantial revisions to the manuscript’s overall structure. Specifically, we subdivided each section into finer-grained subsections based on the meaning of the content and revised paragraphs that previously included multiple topics to ensure that each paragraph focused on a single, coherent theme. Additionally, redundant content—such as repeated descriptions of results in the Discussion section—was removed to clarify the distinct role of each section. Although the total word count has increased owing to the addition of the new analyses described above, we believe that these revisions have clarified the justification of the technical framework, theoretical contributions, and limitations of our study.

3. Clear distinction between human and ML cues

Based on the suggestions provided by both reviewers, we carefully reviewed the entire manuscript and clarified, for each instance, whether the term "cue" referred to kinematic information used by human athletes or that identified as informative for the ML model. Furthermore, to highlight the similarities and differences between these two types of cues, we added new paragraphs to the Discussion section to explicitly address this point. We believe that this distinction helps readers better understand the characteristics of human anticipation skills and the specific limitations of ML models.

We believe that the revisions made in response to the reviewers’ comments have significantly improved the overall quality and clarity of this manuscript. Individual responses to each comment are provided in a point-by-point format in the sections below.

Reviewer #1: This study endeavors to identify the spatio - temporal cues for predicting baseball pitch types and analyze the experience accumulation effect by means of data - driven machine learning approaches. The research perspective holds a certain degree of novelty, and the methodological design is relatively systematic.

Nevertheless:

### 1. Research Sample and Data

- **Limited Sample Size**: The sample consists of only 8 pitchers, all of whom are at the college level. This significantly restricts the generalizability of the research findings. It is advisable to explicitly state this limitation within the discussion section and propose that future research incorporate a more diverse and higher - caliber athlete sample.

Response: As the reviewer pointed out, the limited sample size and uniform skill level of the target pitchers are important limitations of our study. Therefore, based on the reviewer's suggestion, we added a sentence to explicitly mention this point (p. 31, lines 528–531).

- **Insufficient Basis for Pitch Type Classification**: The classification of pitch types is solely predicated on speed differentials (≥10 km/h) and pitchers' self - reports. The absence of objective pitch trajectory data (such as lateral or vertical displacements provided by TrackMan) may introduce classification biases.

Response: Thank you for your comments. Based on the reviewer’s suggestion, we calculated an additional index representing the relative displacement of breaking balls with respect to each pitcher’s fastball using the TrackMan data (p. 11–12, lines 180–190). As shown in the data, the minimum displacement between fastballs and breaking balls was at least 14 cm for all pitchers, which is larger than the approximate diameter of a baseball (~7 cm). This suggests that the pitch trajectories were well separated in our dataset and supports the plausibility of the pitch-type labels.

- **Unbalanced Data**: There is an imbalance in the number of fastballs and off - speed pitches among pitchers. Despite attempts to balance the data during sampling, this may still impact the stability and generalization ability of the model during training.

Response: Thank you for your comment. Based on the reviewer's comments, we also evaluated the influence of the number of datasets and class imbalance in the additional analyses (p. 21, lines 354–355). Although the results were not statistically significant, these parameters showed moderate correlations with the prediction accuracy (p. 24–25, lines 418–420). Therefore, the current dataset construction is fundamentally sound, these factors may still have subtle influences on the results. Accordingly, we have added a note in the revised manuscript to mention this potential influence and encourage further investigation in future studies (p. 29, lines 492–494).

### 2. Methodological Modeling and Interpretation

- **Unverified Association with Human Cognition**: The study highlights the adoption of a logistic regression model due to its interpretability. However, it fails to verify whether the "cues" identified by the model are actually utilized by human batters. It is recommended to clearly demarcate between "machine - identifiable information" and "information practically employed by humans" in the discussion.

Response: We sincerely appreciate the reviewer’s constructive and insightful comment. We agree with the reviewer's opinion that it is necessary to clearly distinguish between information that improves machine learning prediction accuracy and information that is actually utilized by humans. Based on your comments, we have made the following revisions to the manuscript.

First, to avoid potential confusion for readers, we have made consistent clarifications throughout the manuscript. For example, instead of using the generic term “cue,” we now specify the target of the information by using expressions such as “cue for ML” or “cue for human,” where appropriate. Additionally, in the Discussion section, we added paragraphs to discuss the similarities and differences between the cues employed by human athletes and those identified as informative by the ML model (p.26–28, lines 441-470). We believe that this distinction helps clarify the differences for readers and highlights the characteristics of human anticipation skills and the specific limitations of the ML models. Finally, we added a sentence to explicitly state the need for careful future investigation into whether the cues identified by the ML model are accessible and usable by human athletes (p.31, lines 528-531).

We believe that these revisions, which demarcate between " machine-identifiable information" and "information practically employed by humans,” will help readers appropriately interpret the analysis results of this study and avoid overestimating their implications.

- **Assumption of Temporal Independence**: The model is trained independently at each time point, overlooking the temporal dependence inherent in the action sequence. It is suggested to supplement the analysis with sequence models such as LSTM or GRU for comparative purposes and explore the influence of temporal information on prediction.

Response: Based on the reviewer's comments, we conducted an additional analysis in which we implemented the LSTM and GRU models and compared their prediction performance with that of the logistic regression model (p. 19–20, lines 322–330). Consequently, no statistically significant improvement in prediction accuracy was observed (p.24, lines 409–414). This may be because, unlike human observers, the machine learning models in this study could access complete information at each time point without being affected by sensory noise, thereby diminishing the benefit of noise reduction through the temporal accumulation of input information.

We consider these discussions to offer valuable insights into the differences between human and machine cognitive processes. Therefore, we have added the corresponding sentences to the revised manuscript to discuss these points (p. 27–28, lines 464-470).

- **Oversimplified Feature Engineering**: Feature engineering is overly simplistic, relying solely on joint angles while neglecting dynamic features such as joint velocities and accelerations, which may contain more discriminative information.

Response: Based on the reviewer's comments, we conducted an additional analysis to verify the effect of dynamic features on prediction accuracy. Specifically, we trained the logistic regression model using datasets that included joint angular velocity and acceleration in addition to joint angles, and compared the results with those obtained using a dataset containing only joint angle information (p. 19, lines 313–320). As a result, although the difference was not large enough to reach statistical significance given the current sample size, an improvement of approximately 3–5% in the mean prediction accuracy among target pitchers was observed in the later phase of the motion when dynamic features were included (p.24, lines 409-412).

This result suggests that while the current feature engineering is fundamentally sound, these factors may still have subtle effects. Therefore, based on these discussions, we added a sentence mentioning the potential impact of these dynamic features on prediction accuracy (p. 24, lines 412–414).

### 3. Result Analysis and Interpretation

- **Over - Reliance on Cluster Permutation Tests in Statistical Analysis**: Although cluster permutation tests are suitable for time - series data, the failure to report effect sizes (such as Cohen's d) renders it arduous to evaluate the practical significance of the results.

Response: Thank you for your comment. With reference to a previous review paper on the cluster-based permutation test (Meyer et al., 2021), we added information on the effect sizes corresponding to each permutation test result (p. 17–18, lines 292–296). Specifically, we followed the guidance of Meyer et al. (2021) in reporting effect sizes alongside p-values, calculating the maximum effect size within each cluster and the average effect size over the time window encompassing it. The latter tends to yield smaller values than the former, serving as a more conservative estimate or safety margin. However, in some cases, this may result in larger values, depending on the data distribution.

Through these revisions, we believe that the revised manuscript provides a more accurate reporting of the results and greater overall reliability in the analysis.

Meyer M, Lamers D, Kayhan E, Hunnius S, Oostenveld R. Enhancing reproducibility in developmental EEG research: BIDS, cluster-based permutation tests, and effect sizes. Dev Cogn Neurosci. 2021;52: 101036. doi: 10.1016/j.dcn.2021.101036.

- **Inadequate Analysis of Individual Differences**: While individual differences are acknowledged, a deeper analysis of their origins (such as pitching motion styles and pitch type combinations) is lacking. There is a dearth of in - depth exploration regarding why certain pitchers are more predictable.

Response: We sincerely appreciate the reviewer’s constructive and insightful comments. Based on the reviewer’s suggestion, the revised manuscript includes additional analyses aimed at identifying factors contributing to individual differences in prediction accuracy across pitchers and gaining deeper insight into their underlying causes, such as variability in pitching motion, pitch-type combinations, and pitch trajectories (p. 20–21, lines 332–358). Although no statistically significant factors were identified in the analysis, some variables, such as the degree of class imbalance, exhibited moderate correlations with individual prediction accuracy. Therefore, we highlighted these points as potential contributors to individual variability, so that readers can be aware of their possible influence (p. 29, lines 492–494).

Furthermore, we have discussed the potential causes and implications of individual differences more deeply than in the previous version of the manuscript. Overall, we consider that to understand the individual differences in prediction accuracy observed in this study, it may be necessary to conduct more in-depth, individual-specific analyses of pitching motions—such as how each pitcher’s intra- and inter-limb coordination patterns change between fastballs and breaking balls—rather than searching for common explanatory factors across individuals (p. 28–29, lines 472–494).

We believe that these additional analyses and revisions will provide valuable insights for readers regarding how ML analysis can be effectively applied to analyze athletic movements that involve substantial individual variability.

- **Prudent Interpretation of the "Experience Accumulation Effect"**: The improvement in model accuracy with an increase in training samples does not necessarily equate to the "experience accumulation" of human batters. This should be clearly presented as a computational analogy rather than direct evidence of a cognitive mechanism in the discussion.

Response: We agree with the reviewer’s opinion that it is important to clarify that the results of Analysis 2 merely represent a computational analogy or the information accumulation process from the perspective of machine learning models and do not constitute direct evidence of a cognitive mechanism. Based on the reviewer’s comments, we revised the relevant sentences and added a caution to explain this point to the readers (p. 30, lines 509–511).

### 4. Manuscript Writing and Structure

- **Refinement of English Expression**: Some sentences are overly long and exhibit unnatural grammar. It is recommended to seek professional English editing services or the assistance of native speakers for language refinement.

Response: Based on the reviewer's comment, we had the entire manuscript professionally proofread by a professional native English editor. We believe that this revision has improved the clarity and grammatical accuracy of the manuscript and will help readers better understand the content of our manuscript.

- **Chaos in Figure Referencing**: The manuscript frequently refers to "Fig 5–Fig 10", yet some figures in the submitted file are labeled as "In review" or are missing, which severely hinders the review process.

Response: We sincerely apologize for the confusion and inconvenience caused by the missing or mislabeled figure files. Based on the reviewer’s comment, we have carefully reviewed and corrected all figure references and labels to ensure consistency between the main text and the submitted figures. In the revised manuscript, all figures are properly numbered, captioned, and referenced in accordance with journal guidelines.

Additionally, to prevent any similar issues during the review process, we have uploaded a supporting information file that contains a ZIP archive of all original figures along with their captions. If any figure-related issues persist, we kindly ask the reviewers to refer to this supplementary file.

- **Streamlining of the Discussion Section**: Some parts of the discussion section repetitively describe the results. It is advisable to strengthen the comparison with prior research and accentuate the theoretical contributions and practical implications of this study.

Response: Thank you for your comment. Based on the reviewer’s suggestion, we have added the following

---

## [Decision Letter · Decision Letter 1]

6 Feb 2026

A data-driven analysis of spatiotemporal cues and experience accumulation effects for pitch type prediction

PONE-D-25-57679R1

Dear Dr. Takamido,

We’re pleased to inform you that your manuscript has been judged scientifically suitable for publication and will be formally accepted for publication once it meets all outstanding technical requirements.

Kind regards,

Esedullah Akaras

Academic Editor

PLOS One

Additional Editor Comments (optional):

Reviewers' comments:

Reviewer's Responses to Questions

**Comments to the Author**

1. If the authors have adequately addressed your comments raised in a previous round of review and you feel that this manuscript is now acceptable for publication, you may indicate that here to bypass the “Comments to the Author” section, enter your conflict of interest statement in the “Confidential to Editor” section, and submit your "Accept" recommendation.

Reviewer #1: All comments have been addressed

Reviewer #2: All comments have been addressed

2. Is the manuscript technically sound, and do the data support the conclusions?

Reviewer #1: Yes

Reviewer #2: Yes

3. Has the statistical analysis been performed appropriately and rigorously? 

Reviewer #1: Yes

Reviewer #2: Yes

4. Have the authors made all data underlying the findings in their manuscript fully available?

Reviewer #1: Yes

Reviewer #2: Yes

5. Is the manuscript presented in an intelligible fashion and written in standard English?

Reviewer #1: Yes

Reviewer #2: Yes

6. Review Comments to the Author

Reviewer #1: I appreciate the author's revisions; I now believe the article is ready for publication in a journal.

Reviewer #2: I thank the authors for their detailed responses and rigorous revisions to the manuscript. The changes made reflect real attention to the comments made during the review process and demonstrate a sustained effort to clarify and strengthen the study's contribution. The reformulation of several conventional methods, clarification of methodological challenges related to small data sets and interindividual variation, and the restructuring of the Introduction, Materials and Methods, and Discussion sections have significantly improved the readability and coherence of the argument. In particular, the thematic segmentation of the Discussion and the clearer delineation of the practical implications and limitations of machine learning models contribute to a more balanced and nuanced interpretation of the results. I believe that these revisions make a substantial contribution to the scientific quality of the article and will facilitate the development of its contribution by readers. I congratulate the team of authors for the careful and professional way in which they approached the review process and for the care they need, which undoubtedly reinforce the value of the manuscript.

7. PLOS authors have the option to publish the peer review history of their article (what does this mean?). If published, this will include your full peer review and any attached files.

Reviewer #1: No

Reviewer #2: **Yes:**Ilie Eva

---

## [Editor Report · Acceptance letter]

PONE-D-25-57679R1

PLOS One

Dear Dr. Takamido,

I'm pleased to inform you that your manuscript has been deemed suitable for publication in PLOS One. Congratulations! Your manuscript is now being handed over to our production team.

Kind regards,

on behalf of

Dr. Esedullah Akaras

Academic Editor

PLOS One